# Synthesizing Recent Trends in Interventions and Key Ecosystem Services in Indonesian Peatland

Hyun-Ah Choi [1,2], Cholho Song [1,*], Chul-Hee Lim [3,4,*], Woo-Kyun Lee [1,5], Hyunyoung Yang [6] and Raehyun Kim [6]

1   OJEong Resilience Institute, Korea University, Seoul 02841, Republic of Korea;
    sosobut.choi@gmail.com (H.-A.C.); leewk@korea.ac.kr (W.-K.L.)
2   Hanns Seidel Foundation Korea Office, Seoul 04419, Republic of Korea
3   College of General Education, Kookmin University, Seoul 02707, Republic of Korea
4   Department of Forestry, Environment and Systems, Kookmin University, Seoul 02707, Republic of Korea
5   Division of Environmental Science and Ecological Engineering, Korea University,
    Seoul 02841, Republic of Korea
6   Global Forestry Division, Future Forest Strategy Department, National Institute of Forest Science,
    Seoul 02455, Republic of Korea; hyhy0672@korea.kr (H.Y.); rhkim@korea.kr (R.K.)
*   Correspondence: cholhosong@korea.ac.kr (C.S.); clim@kookmin.ac.kr (C.-H.L.)

**Abstract:** This study conducted a systematic literature review focusing on peatlands studies in Southeast Asia, specifically in Jambi, South Sumatra, and the Riau province of Indonesia, covering the period from 2001 to 2023. To ensure the quality and rigor of the analyzed articles, a critical process and systematic review were employed. Journal articles were extracted using reputable resources, including Google Scholar and Scopus, to enhance the validity and reliability of the research results. We identified significant research topics based on region, province, and sector. Additionally, we synthesized the existing classification of ecosystem services, drawing on previous studies conducted in Indonesia. These services were categorized as provisioning, regulating, cultural, and supporting services. We also reviewed the classification of ecosystem service types based on peatland degradation and restoration. This study identified evidence of peatland intervention to evaluate ecosystem services in Indonesia. We found that large-scale cultivation and production of palm oil, local policies, and forest fires were the main intervening factors in Indonesian peatlands. Furthermore, Indonesian peatlands have undergone conversion to oil palm, timber, and crop plantations. It is imperative to substantiate the effectiveness of future peatland restoration plans and further refine the quantification of services provided by peatland ecosystems through cooperative projects.

**Keywords:** peatland; systematic review; interventions; ecosystem services

## 1. Introduction

Land cover and use changes are critical drivers of carbon emissions and loss of ecosystem services [1–3]. Understanding degradation drivers and their relationship with ecosystem services is necessary to secure land degradation neutrality (LDN) as well as enhance the ecological nexus of the water–food–energy (WFE) system [2,4]. LDN is defined as "a state whereby the amount and quality of land resources necessary to support ecosystem functions and services and enhance food security remain stable or increase within specified temporal and spatial scales and ecosystems" [5]. LDN enhances land and ecosystem-based natural capital and its associated ecosystem services [6]. LDN and the WFE system are inextricably linked. LDN provides multiple services and maintains ecosystem functions upon which water–food–energy provisioning ultimately depends.

Peatlands are being impacted by land cover change and climate change disturbances [7]; LDN assessments should look at the balance of ecosystem services and the WFE system. Peatlands have the global potential for greenhouse gas (GHG) emission reductions through

conservation and management. Peatlands are source of organic carbon transfer to surface water at the watershed scale [8,9] and sequester 0.37 Gt of $CO_2$ a year [10]. Peat soils contain more than 600 Gt of carbon, which exceeds the carbon stored in all other vegetation types including the world's forests [10,11]. Peatlands are the most extensive terrestrial organic carbon stock, and the conservation and protection of peatlands are critical to climate change. However, poorly implemented peatland management can result in land degradation and habitat loss. Wetland management, including peatland management, is progressively adopting the ecosystem services concept and valuation methods for peatlands [12]. The contribution of peatlands and their sustainable use are essential because they provide a range of ecosystem services. The peatland ecosystem provides livelihoods for local communities through agricultural cultivation and fisheries, both of which are conducted in accordance with local wisdom [13]. Peatlands can be classified on the basis of their water source [14] and peat composed of fabric peat, capric peat, and mineral soil [15]. Even though we know the importance of peatlands and their protection, they face ongoing pressure that leads to their reduction. The reduction in peatland is attributed to tropical peat fires occurring on sites disturbed by logging, various types of agriculture, road construction, and abandoned post-agriculture areas. These disturbances make traditional classification schemes less applicable [15,16].

Indonesia holds 36% of the world's tropical peatland, making it the largest country in the tropics in this regard [17,18]. For example, Sumatra is part of the Sundaland biodiversity hotspot [19]. However, the peatlands in Indonesia face severe challenges due to the recent expansion of agriculture, resource production, and natural disasters, including wildfires in Indonesia [20–22]. The diminishing extent of Indonesian peatlands can have a significant impact on climate change and ecosystem management. Simultaneously, prioritizing peatland restoration has emerged as a crucial environmental strategy in addressing the climate change crisis. Indonesia's expansive peatlands play a crucial role in global carbon sequestration, biodiversity, and water resources [18,23–25]. However, unsustainable practices such as drainage for agriculture and fires have led to the degradation of these ecosystems, significantly contributing to GHG emissions [26–28]. To address this issue, various interventions have been implemented, but their effectiveness depends on a complex interplay of factors.

International attention to Indonesia's peatlands began to rise in 2015 following a massive forest fire in Londerang, Jambi, Sumatra, Indonesia [29–31]. The haze resulting from these fires led to smog in neighboring countries, including Singapore and Malaysia, and escalated into an international issue. Subsequently, restoration projects have tapped into international funds to conduct primary surveys, and restoration directions have also been proposed [32]. Recently, collaborative peatland projects in Indonesia have witnessed growth. One notable example is the Sustainable Community-based Reforestation and Enterprises (SCORE) project in partnership with the Center for International Forestry Research and World Agroforestry (CIFOR-ICRAF) and the National Institute of Forest Science, which focuses on peatland restoration [33]. It aims to enable long-term mitigation of GHG emissions through climate-smart agroforestry approaches, which increase carbon sinks, biodiversity and ecosystem services through landscape restoration and enhance local income through markets for sustainable timber and agri-food and fishes, essential oils, and biomass enterprises [33]. Nevertheless, most restoration projects have primarily focused on Kalimantan, with relatively few studies conducted in Sumatra, despite the significant distribution of peatlands in the region. Furthermore, there has been limited research and discussion on peatland carbon. The most crucial element of peatland restoration is rewetting, which is the process of transforming degraded peatlands back into wetlands. This necessitates concerted efforts to restore wetlands and rainforests by replanting native plants for sustainable management.

As an initial step in the SCORE collaborative research, we conducted a systematic review of major studies related to peatlands. This study aimed to identify research trends in peatlands in the Kalimantan and the entire Southeast Asia regions while reviewing

ecosystem services pertinent to SCORE collaborative research sites. As climate change accelerates and natural ecosystems face continued degradation, the importance of peatlands is garnering increasing attention. Peatlands serve as significant carbon stores and offer various ecosystem services. However, they are subject to substantial pressure for degradation resulting from the income of local communities, such as palm oil and crops. Therefore, appropriate understanding of the current status of land degradation focusing on research areas is necessary for developing land management policies and plans. Furthermore, clarification of linked ecosystem services is required to understand the complex degradation and restoration schemes in these areas.

In this study, we aimed to evaluate the ecosystem services of peatlands in the provinces of South Sumatra and Jambi in Indonesia, considering various land use changes. The goal is to provide a basis for developing sustainable strategies for peatland restoration and use. This study seeks to identify, assess, and synthesize evidence on interventions in peatlands to evaluate ecosystem services in Indonesia. The goal of this study is to facilitate the use of this evidence in informing policy and practice decisions, especially in peatland management. To achieve these objectives, we address the following questions: (1) What types of ecosystem services are included in peatlands in Indonesia? (2) What factors influence the effectiveness of peatland ecosystem management interventions in Indonesia?

## 2. Materials and Methods

We conducted a systematic literature review using morphological analysis. This study involved searching the Google Scholar database, the International Initiative for Impact Evaluation: 3ie Development Evidence Portal, and Scopus for research related to peatland, agriculture, fishing, and forestry in Indonesia. The specific study sites included Perigi, South Sumatra, Indonesia and Londerang, Jambi, Indonesia, focusing on peatland and restoration research conducted between January 2001 and July 2023 ($n$ = 1196) (Table 1). We used the preferred reporting items for systematic reviews (PRISMA) flow [34] to assist in the identification and selection of articles on search platforms (Figure 1).

**Table 1.** Number of studies in our systematic review.

| Categories | | Contents | No. of Studies |
|---|---|---|---|
| Region | SE Asia | Southeast Asia, ASEAN | 35 |
| | Sumatra | Sumatra, Jambi, South Sumatra, West Sumatra, East Sumatra, Riau | 131 |
| | Kalimantan | Kalimantan, cities and parks in Kalimantan, Borneo | 180 |
| Province | Jambi | Jambi province | 21 |
| | S. Sumatra | South Sumatra, Sumatra Selatan province | 32 |
| | Riau | Riau province | 87 |
| Sectors | Air | Greenhouse gas, emission, carbon dioxide, aerosol, emission, $CO_2$ | 103 |
| | Fire | Fire and burning | 222 |
| | Wildfire | Wildfire | 9 |
| | Palm oil | Palm oil and oil palm | 75 |
| | Plantation | Plantation and cultivation | 60 |
| | Restoration | Restoration | 72 |
| | Degraded | Degradation and deforestation | 57 |
| | Diversity | Species and diversity | 35 |
| | Soil | Soil, carbon stock | 77 |
| | Total | | 1196 |

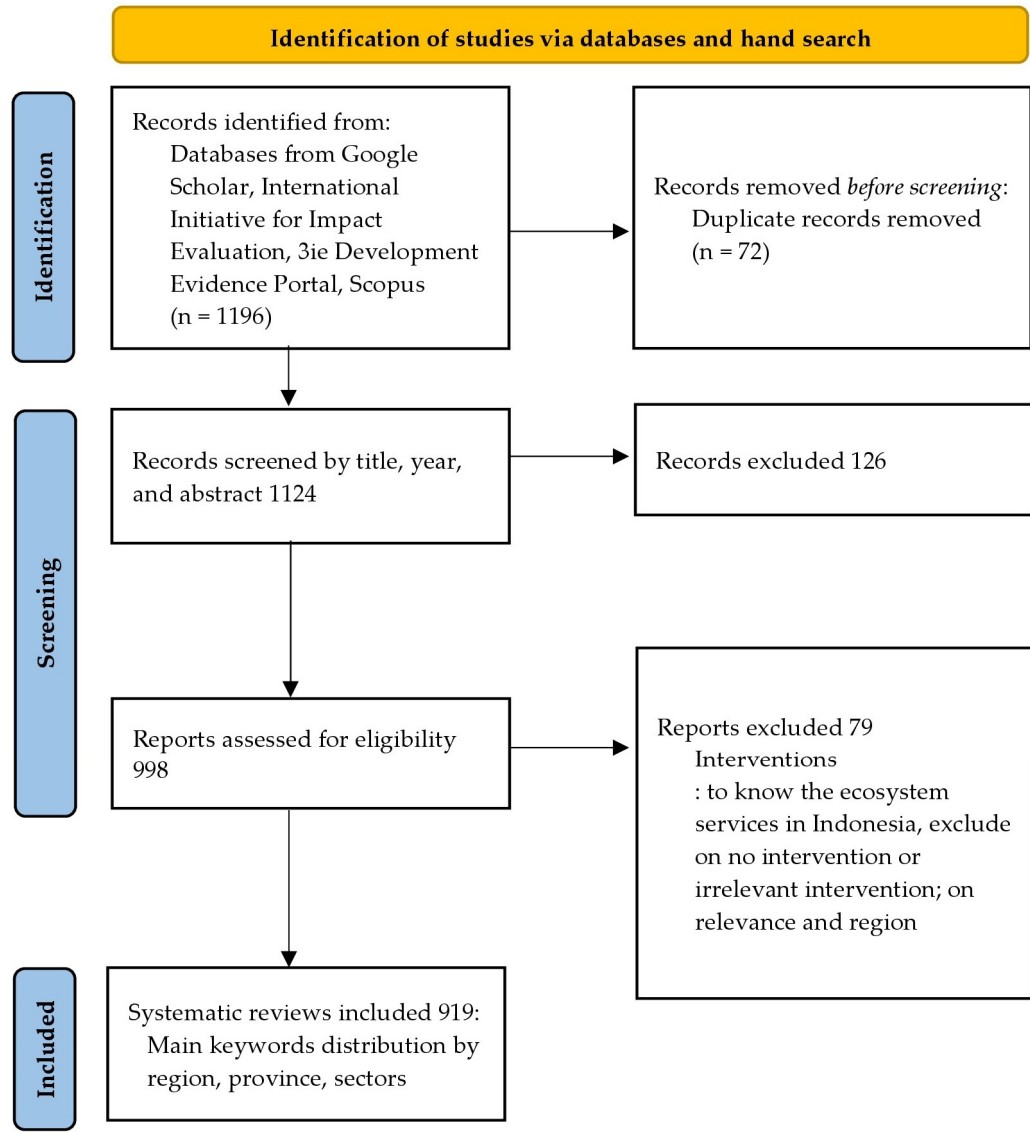

**Figure 1.** Study framework of the included systematic review using PRISMA flow.

In the second phase, we categorized the searched literature using morphological analysis. The morphological analysis was conducted based on Python 3.10.12. Based on the selected literature found by searching in the previous step, data were collected through crawling, and the characteristics of words and morphemes were analyzed based on packages such as pandas 1.5.3, wordcloud 1.9.2, selenium 4.14.0, nltk 3.8.1, counter, and string. Through this analysis, we identified the most frequently repeated words from the general reviews except particles and articles. This morphological analysis was regionally, provincially, and sectorally categorized, and the main contents were identified based on previous studies (Table 1). Building upon previous studies on ecosystem services in Indonesia and encompassing the study sites, we synthesized existing ecosystem service type classifications as provisioning, regulating, cultural, and support services. Following a review of the classification of ecosystem service types considering peatland degradation and restoration, we ultimately reclassified ecosystem functions and services according to the Millennium Ecosystem Assessment (MA), which assessed the consequences of ecosystem change for human well-being classification system to align with peatland assessment. In MA [35], provisioning services encompass tangible outputs such as food, water, and energy obtained from natural resources. In contrast, regulating services encompass natural phenomena such as cycling–absorbing carbon and GHG, purifying water, and preventing

disasters. Supporting services constitute natural phenomena necessary for the operation and maintenance of an ecosystem, such as nutrient cycling or soil production. Cultural services encompass spiritual support from nature, including ecotourism and recreation, and the influence of nature on human culture. According to Karki et al. [36], assessments conducted in the Asia–Pacific region have adopted MA as a framework. However, most assessments have focused on forests, coasts, and croplands, with a lack of evaluations for wetlands, islands, cities, and dryland ecosystems.

This study adopted specific types of intervention structures and clearly defined its scope. Interventions in peatlands in Indonesia encompass measures that aim to enhance resilience and adaptive capacity, thereby directly impacting the achievement of a majority of the UN's Sustainable Development Goals. Only selected peatland intervention types were included, and these selections were informed by empirical research from previous studies on peatlands. Specifically, these are categorized into eight groups: nature-based solutions, which arise from avoiding the conversion of natural areas, reducing the impact of disturbance on ecosystem GHG emissions through better management and ecological restoration [11,37]; ecosystem-based wetland management, which aims to protect and enhance the sustainability, diversity, and productivity of wetland [38]; peatland management; restoration; agroforestry; built infrastructure; peat wetting infrastructure; and rehabilitation. Additionally, this study reviewed peatland ecosystem service assessments, suggesting an examination of the potential impact on ecosystem services and peatland conservation, adapted from Burkhard et al. [24], Choi et al. [39], and Maes et al. [40].

## 3. Results

### 3.1. Publication Trends over Time

Individual screening was conducted for the studies, leading to the exclusion of most studies because of non-relevance to the topic. A total of 919 studies were reviewed (Supplementary Table S1). We excluded 277 studies for reasons such as lack of intervention or the presence of an intervention that did not meet the inclusion criteria, including study site, duplicates, relevance, and study design. Following the screening of all records at the title and abstract stage, we reviewed the articles to assess eligibility. The highest volume of research was conducted in 2021, with 149 studies, followed by 130 in 2020, and 120 in 2022 (Figure 2). This was in response to Indonesia's 2015 tropical forest fires, which caused diplomatic tensions in Singapore and Malaysia [41].

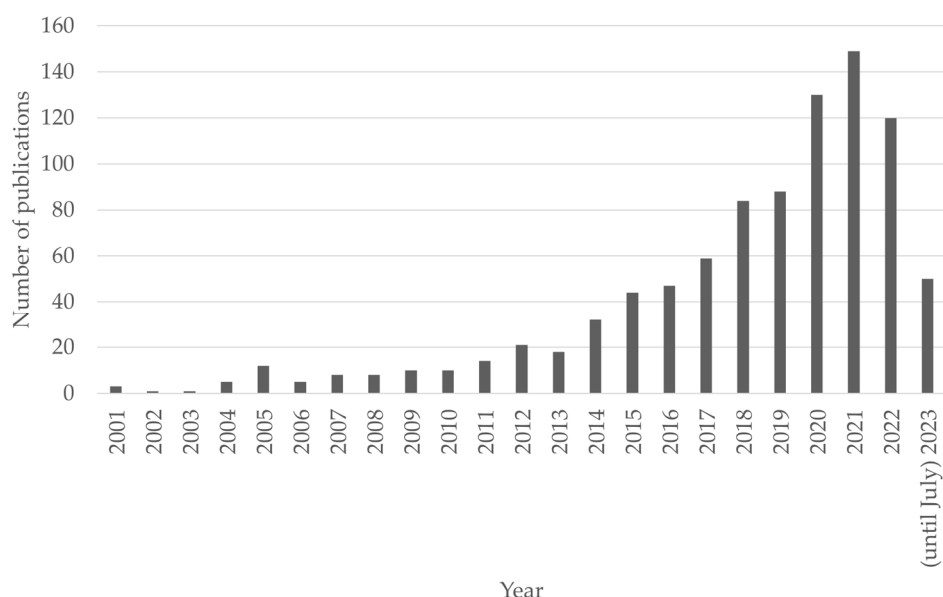

**Figure 2.** Number of publications on Indonesian peatland in international peer-reviewed scientific papers from January 2001 to July 2023.

### 3.2. Distribution of Studies by Region

By region, research in Southeast Asia is predominantly centered in Indonesia, where peatlands and forest fires are the most prevalent subjects. In Sumatra and Kalimantan, Indonesia, peatlands and forest fires were the most common topics of research. However, in Kalimantan, there was a greater focus on agriculture, vegetation, and conservation. Unlike the other regions, the Kalimantan region recorded a high frequency of conservation terms, with 33 recorded. In contrast, in Sumatra, research topics such as forests, soils, carbon, and restoration were prominent (Figure 3).

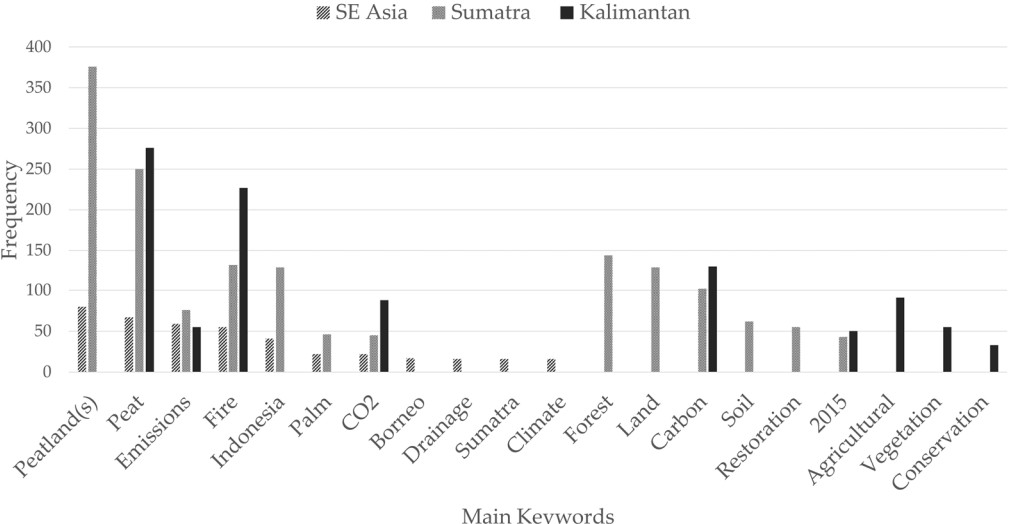

**Figure 3.** Distribution of main keywords by region: Southeast Asia, Sumatra, Kalimantan from January 2001 to July 2023.

### 3.3. Distribution of Studies by Provincial Level

In Jambi Province, the main keywords were palm oil, restoration, swamp, plantation, degradation, and agroforestry, while in Riau Province, the main keywords were peatland, palm oil, carbon, data, species, soil, management, and restoration. In South Sumatra, the main keywords were peatlands, peat, land, fires, community, rainfall, and restoration (Figure 4). At the provincial level, conservation or protection scored very low in the main keywords analysis. Jambi province recorded 4 times in conservation and 2 in protection, while Riau province recorded 2 points in conservation and 6 in protection. South Sumatra recorded only 2 points in conservation. This showed that peatland conversion and restoration of degraded areas were considered more important than conservation and protection. It is also likely that previous studies focused on restoration were prioritized in order to better conserve degraded areas.

The primary threats to Indonesia's peatlands are large-scale palm oil cultivation and production, local resettlement policies, and catastrophic forest fires [42,43]. As analyzed in previous studies, Indonesia's peatlands have undergone conversion to oil palm, timber, and crop plantations [7,43]. The establishment of canals for agriculture has contributed to the drying of peat soils, escalating the frequency and severity of wildfires. Peatlands in South Sumatra are recognized as constituting 20% of Indonesia's total peatlands and 10% of the world's peatlands, with substantial peatland areas also present in the neighboring Jambi province [44]. In recent years, the degradation of peatlands in Indonesia, driven by the expansion of agriculture and plantations, has been evidenced to result in diverse forms of ecosystem service degradation. These include logging, thermal power plants, and road networks [45]. Simultaneously, there is an interest in sustainable community-based peatland restoration on degraded peatlands, and various projects are ongoing.

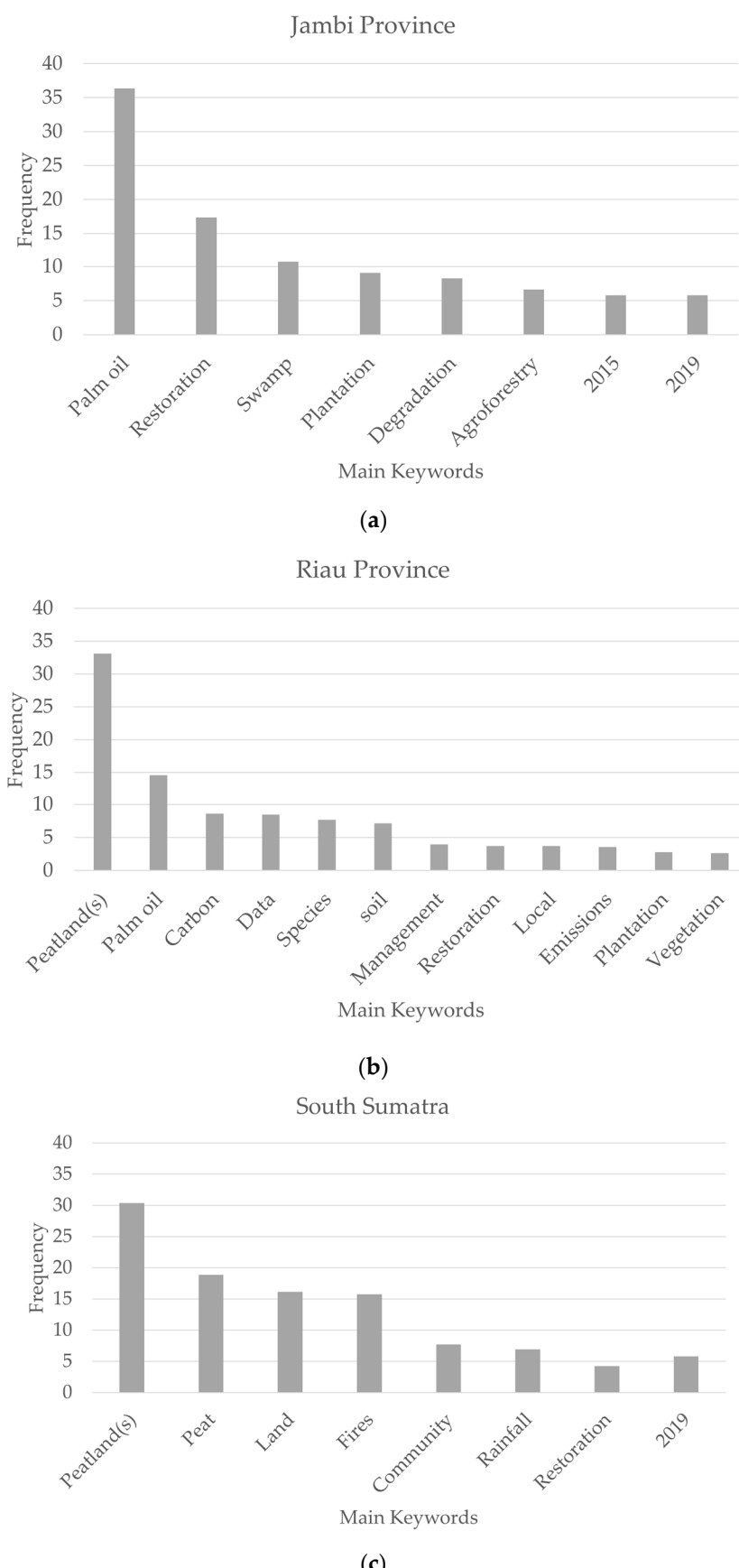

**Figure 4.** Distribution of main keywords by provinces; (**a**) Jambi, (**b**) Riau, and (**c**) South Sumatra from January 2001 to July 2023.

### 3.4. Distribution of Studies by Sectors

In the past, tropical peatlands were converted to oil palms and other plantations for exploitation and utilization, with more extensive research on these areas compared with other topics. However, recent restoration studies have increasingly focused on peatland rewetting. Various studies on tropical forest fires and forest fires, as well as many studies on GHG emissions such as those of carbon dioxide and methane, have been conducted in different sectors (Figure 5). The ongoing degradation of tropical peatlands is more advanced in terms of research on land reclamation and restoration compared with other areas. In some instances, new land has been cleared for agriculture, necessitating additional research on land reclamation beyond existing degraded areas. Similar to the regional and provincial levels, the sectoral analysis also showed that previous studies focused on restoration, rather than conservation and protection, have been prioritized to better preserve degraded peatland areas.

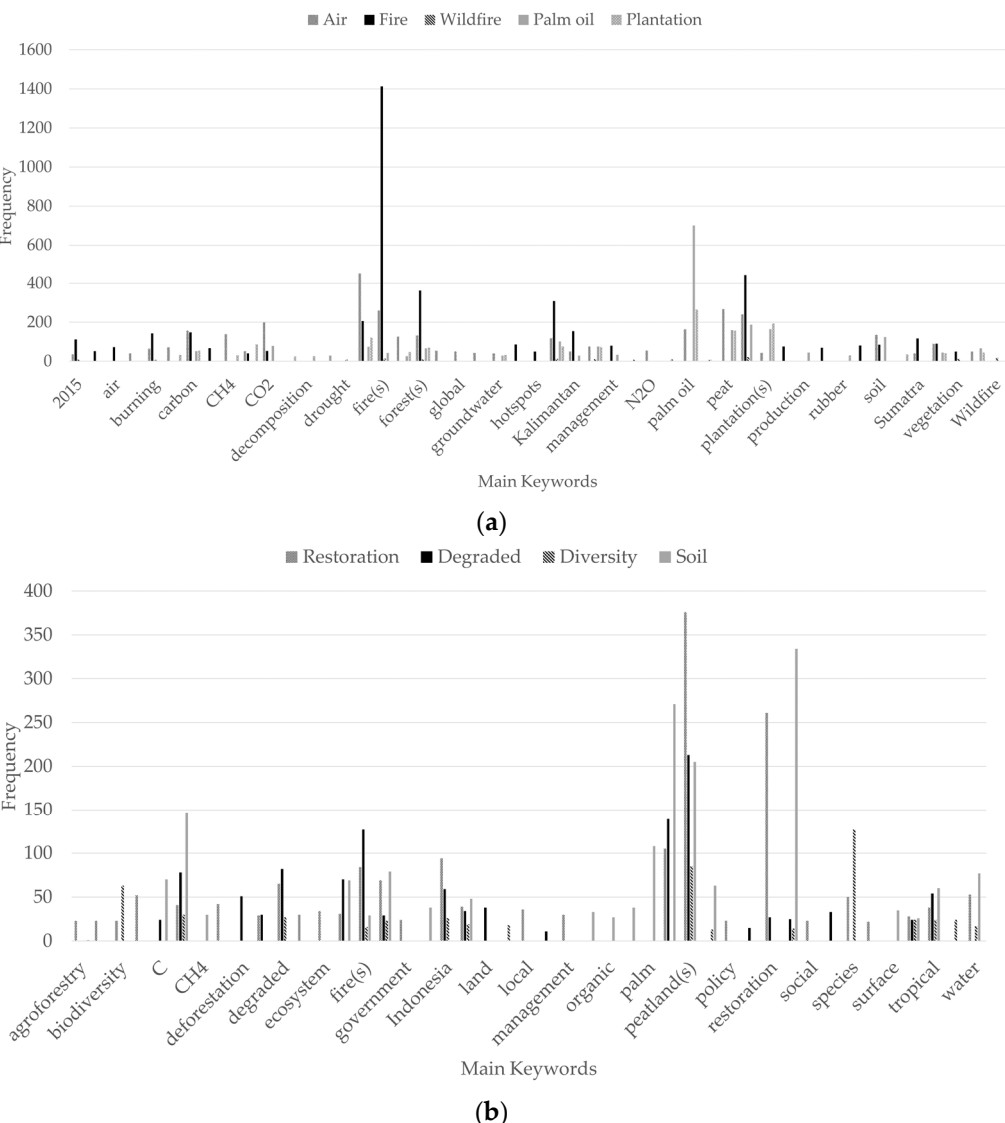

**Figure 5.** Distribution of main keywords by sectors; (**a**) air, fire, wildlife, palm oil, and plantation; and (**b**) restoration, degraded, diversity, and soil from January 2001 to July 2023.

### 3.5. ES Interventions in the Systematic Review

Based on previous studies on tropical forests and peatlands, we selected ecosystem service functions and services that are meaningful to the study site. We also classified the peatland ecosystem services in Table 2 through literature reviews and discussions with

experts involved in the SCORE initiative. The selected peatland ecosystem services have been assessed and quantified in existing studies. The most frequently and consistently crucial ecosystem services were the regulating services of air quality control, global and regional climate, and water regulation, along with cultural services such as cultural heritage, recreation, and tourism.

**Table 2.** Selected ecosystem services from international peer-reviewed scientific papers in Indonesia (South Sumatra, Jambi, Riau province).

| Ecosystem Services | | | |
| --- | --- | --- | --- |
| **Provisioning** | **Regulating** | **Cultural** | **Supporting** |
| Energy (biomass) | Climate regulation/carbon storage | Recreational and tourism | Biodiversity/habitat |
| Food (Non-wood forest products) | Water regulation | Spiritual and inspirational | Soil formation |
| Water supply | Water purification | Educational | Nutrient cycling |
| Peat substrate | Erosion protection | Cultural heritage value | Soil fertility |
| Traditional agriculture | Natural hazards | Social Relations/Local Community | |
| Livestock grazing | Air quality | | |
| Material extraction (rubber, grass) | | | |
| Paludiculture | | | |
| Agro-silvo-fishery | | | |

However, the Indonesian peatland area decreased mainly due to the advanced use of the peatland as a cultivation area and due to land cleaning [46,47]; protection and conservation efforts are not sufficient [48–50]. Indonesian peatlands provide land for plantations and cultivation of palm oil, rubber, and rice, and this affects peatlands' deforestation and biodiversity loss [50–52]. Therefore, the main research topic was the impact of peatland wildfires on climate change and peatland degradation and restoration. We found that in Indonesia's peatlands, large-scale cultivation and production of palm oil, local resettlement policies, and catastrophic forest fires were the main intervening factors. It has been observed that Indonesia's peatlands have undergone conversion into oil palm, timber, and crop plantations. With regard to peatland degradation, soil moisture, groundwater levels, weather conditions, carbon storage, and human impact were the main topics. The key features of peatland restoration include the benefits of peatland restoration, forest distribution, species diversity, and carbon emission factors because the restoration is based on interconnections between nature and people and is open to multiple interaction possibilities. On the other hand, the construction of waterways for agriculture has resulted in the drying of peat soils and increased the frequency and severity of wildfires. In addition, rewetting, paludiculture, agro-silvo-fishery, and environmental potential after rewetting were considered the intervention factors, which included peatland degradation, restoration, and the current ecological status of the Indonesian peatlands (Table 3).

**Table 3.** Intervention and related activities in international peer-reviewed scientific papers in Indonesia.

| Interventions | Related Activities |
| --- | --- |
| People | Palm oil plantation, cultivation, and production; fires |
| Rehabilitation | Forests, agro-silvo-fishery; paludiculture |
| Infrastructure | Built infrastructure; peat wetting infrastructure |
| Ecosystem biodiversity | Improving land management and protection |

## 4. Discussion

To answer the first research question, the peatland ecosystem services value assessment included provisioning services such as energy, water, peat substrate, paludiculture, agro-silvo-fishery; regulating services such as climate regulation, water regulation, and natural hazards; cultural services such as cultural heritage values, social relations, and local community; and supporting services such as biodiversity and soil fertility related to LDN and WFE systems. As an answer to the second research question, the major factors that influence the effectiveness of peatland ecosystem management interventions in Indonesia were people, rehabilitation, infrastructure, ecosystem and biodiversity. Recently, studies

have shown that keywords such as restoration and ecosystem significantly increased in frequency. Integrated ecosystem restoration and utilization with long-term aftercare are well established in Indonesia. The government of Indonesia also established a national-level peatland restoration agency and designed a management plan to mitigate climate change [53]. Clarke and Rieley [54] stated that peatland restoration should be implemented through cooperation, coordination, and governance with local communities Furthermore, it is essential to assess the sufficiency and sustainability of post-restoration management methods and whether there is an appropriate support system through consultation with local communities. For Indonesia, it is crucial to advocate for a system in which the government and local communities engage in agreements with landowners in peatland restoration areas. These agreements should support activities aimed at preserving and enhancing ecosystem services and providing appropriate compensation to landowners. Our results also support this restoration direction. Peatlands are changing over time due to people's utilization. It is necessary to emphasize the benefits of peatland conservation for local communities at local, national and global levels.

Peatlands are increasingly recognized in the environmental field as a means to respond to climate change. In terms of climate action, at the 27th Conference of the Parties (COP) of the United Nations Framework Convention on Climate Change, the protection and restoration of water, marine, agricultural, and forest ecosystems were among the topics discussed. Partial technical guidelines for the practical implementation of Article 6 (the International Carbon Market) of the Paris Agreement, which was established at COP 26, were also adopted [55]. In addition, at COP 26, the Peatland Pavilion was held for the first time, and discussions on the various services provided by peatlands and their assessment are ongoing [56]. The transition from a forest-focused debate to the importance of wetlands, including peatlands, as carbon sinks means that discussions and related agendas will be developed. To address the issue of "loss and damage" resulting from the negative effects of climate change, a fund was established at COP 27. This fund places an emphasis on nature-based solutions, WFE systems, reducing emissions from deforestation and forest degradation in developing countries, and actively protecting environmental ecosystems and restoring degraded ecosystems through sustainable forest conservation and management. Various ecosystem services are expected to be assessed in Indonesia under this restoration perspective, including reducing emissions from deforestation, preventing the degradation of forests and peatlands, promoting sustainable forest management, and generating extra benefits (Figure 6).

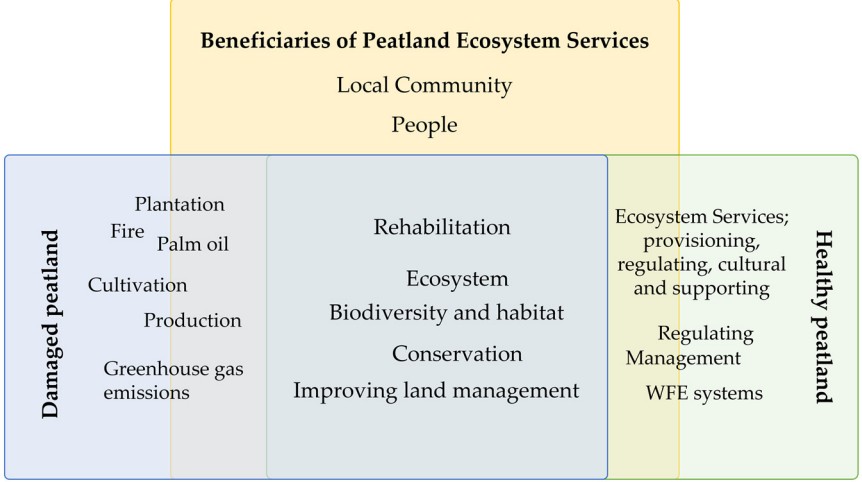

**Figure 6.** Considerations for peatland ecosystem services aligning conservation and sustainable management.

In this study, we reviewed the overall peatland issues in Indonesia and identified the recent keywords and topics attracting the most attention in Indonesian peatland re-

search at regional, provincial and sectoral levels. The literature review shows that it is restoration, rather than conservation and protection, that has been prioritized to better preserve degraded peatland areas in Indonesia. This result reflected a relative change in research interest in peatland issues in Indonesia. There were noticeable keywords, such as restoration, fire, land, and palm oil, that consistently maintained their high ranking together. There were noticeable keywords, such as restoration, fire, land, and palm oil (and the years 2015 and 2019) that consistently maintained their high ranking together, suggesting that peatland restoration has attracted more interest in Indonesian peatland research after peatland has been destroyed. Official Development Assistance projects aimed at peatlands in Indonesia are currently being implemented in various ways, such as afforestation and reforestation projects in peatlands and the prevention of surrounding forest degradation as restoration. In particular, different community forestry strategies, such as community forests, are being implemented in Indonesia; therefore, qualitative and quantitative evaluation criteria for ecosystem services targeting Indonesian peatlands should be applied. As a result, various ecosystem service elements should be used as evaluation criteria to evaluate the effectiveness of various forestry projects targeting rainforests and peatlands in Indonesia [57,58]. This study will serve as a basis for the selection of leading peatland services.

This study also identified evidence of intervention in peatlands to evaluate ecosystem services in Indonesia. It is possible to support activities aimed at protecting and restoring peatlands in terms of biodiversity and carbon absorption, and it is likely that researchers will propose projects that can significantly improve the income of local communities by preventing forest conversion and carrying out peatland protection. Ecosystem service assessment is needed at the national level to support policy makers based on ecosystem function, ecological homogeneity, and anthropogenic land cover and use. However, there is a limitation that the main keywords can only be inferred by relying on researchers' knowledge and experts involved in the SCORE initiative; cross-examination by researchers, including practical field-based assessment and spatio-temporal analysis, is required to overcome this limitation. For further analysis, if quantitative research such as data analysis can be conducted in parallel to explain the changes to main keywords and interventions in detail, it will be possible to obtain a deeper understanding of the status of Indonesian peatlands.

## 5. Conclusions

As climate change intensifies and natural ecosystems face mounting degradation, the crucial role of peatlands becomes increasingly important. These vast carbon stores offer a diversity of ecosystem services, yet they face immense pressure of being degraded for the income of local communities from activities such as palm oil and crop cultivation. This study employed a systematic review of key peatland-related studies to explore both research trends in Indonesia and Southeast Asia and review ecosystem services within SCORE collaborative research sites. We found that peatland research focused on restoration, rather than conservation and protection, has been prioritized to better preserve degraded peatland areas. In Indonesia's peatlands, large-scale cultivation and production of palm oil, local resettlement policies, and forest fires were the main intervening factors. It has been observed that Indonesia's peatlands have undergone conversion into oil palm, timber, and crop plantations.

Through international or cooperative projects, it is necessary to demonstrate the efficacy of peatland restoration in the future and to further refine the quantification of the services that peatland ecosystems provide. For instance, a preliminary feasibility study on the application of quantification assessment to Indonesian peatlands with the potential to be used as palm oil plantations could be conducted. Furthermore, collaboration in peatland conservation can be anticipated through the engagement of diverse stakeholders in the peatland domain, including the government, local communities, and researchers. Conversely, realizing the value of ecosystem services requires various forms of technical support. Furthermore, field data collection, growth observation, and monitoring are

essential to precisely analyze the peatland ecosystem services presented in this study, and it is imperative to consider linking them to future climate change inventories and other carbon credit projects.

**Supplementary Materials:** The following supporting information can be downloaded at: https://www.mdpi.com/article/10.3390/land13030355/s1, Table S1: The list of selected articles for systematic review.

**Author Contributions:** H.-A.C., C.S. and C.-H.L. carried out the conceptualization, methodology, analysis, data curation, and manuscript writing, and obtained funding. W.-K.L., H.Y. and R.K. carried out conceptualization, review and editing. All authors have read and agreed to the published version of the manuscript.

**Funding:** This research was funded by OJEong Resilience Institute (OJERI) at Korea University as Basic Science Research Program through the National Research Foundation of Korea, the Ministry of Education (NRF-2021R1A6A1A10045235) and Kookmin University grant.

**Data Availability Statement:** The original contributions presented in the study are included in the article/supplementary material, further inquiries can be directed to the corresponding author.

**Acknowledgments:** We want to thank all colleagues who contributed to this study's development and the National Institute of Forest Science. We are also grateful for the comments of anonymous reviewers and help from Sanghyo Moon and and Jisang Lee.

**Conflicts of Interest:** The authors declare no conflicts of interest.

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
