# Peer review of "Synthesizing Recent Trends in Interventions and Key Ecosystem Services in Indonesian Peatland"

_land, doi:10.3390/land13030355_

Round 1
Reviewer 1 Report
Comments and Suggestions for Authors## General Comments
- The paper provides a good overview of research trends and ecosystem services related to peatlands in Indonesia. The systematic literature review methodology is sound.
- The paper is well-structured and covers the key aspects of interventions, ecosystem services, and research gaps related to Indonesian peatlands. The figures and tables effectively summarize key information.
- The research questions are relevant and align well with the systematic review methodology. The conclusions directly address the research questions.
- The paper could benefit from some revisions to improve clarity and flow in certain sections. Additionally, expanding on certain concepts and definitions introduced would make the paper more accessible.
## Specific Comments
- In the introduction, provide more background on the importance of peatlands globally and definitions of key terminology (e.g. land degradation neutrality, ecological nexus).
- In the methods section, consider explaining the search strategy, screening criteria, and data extraction process in more detail. A PRISMA flow diagram could help visualize the screening process.
- For Figures 2-4, increase font size of words in word clouds and provide more interpretation of key terms in captions.
- In the results, provide more explanation when introducing new concepts (e.g. nature-based solutions, ecosystem-based wetland management). Define acronyms like MA, LDN, WFE.
- In the discussion/conclusion, summarize key findings by research question and emphasize implications for policy and practice. Limitations of the systematic review could also be noted.
Author Response
We appreciate your valuable comments and suggestions for improving the manuscript. We have revised the manuscript. Please see the attachment.

Reviewer 2 Report
Comments and Suggestions for Authors
The manuscript entitled “Evidence Review of Interventions in Indonesia Peatland for Ecosystem Services Assessment” focused on evaluating the intervening elements and assessment of ecosystem services of peatland ecosystems. Considering the role of peatlands especially in Indonesia, in carbon sequestration, biodiversity and water resources, their conservation is of utmost importance. The study is novel in terms of land degradation neutrality and climate change, but it fails to provide a systematic and comprehensive review. The major concerns are listed below:
1. The title is not particularly appealing; I recommend that the authors replace it with one that is simple but informative.
2. I found the abstract to be very informative and interesting.
3. The introduction section is adequate and provides useful background information. However, a few sentences i.e. line no. 45-46, “According to Wust and Bustin…”this sentence seems confusing, kindly rewrite it.
4. Material and methods: provide a different caption for Figure 1.
5. Results and discussion:
· Ensure that all figures are numbered in a sequence.
· The major concern is that the results are not enough to satisfy the proposed objectives. The results section mainly emphasised on different keywords used and studies conducted. Therefore, I don’t understand the real outcome of this study.
· It is suggested to present the concrete findings of this review rather than discussing the already published data.
Author Response

(The authors gave the same response as above.)

Reviewer 3 Report
Comments and Suggestions for Authors
The authors carry out a review of the state of conservation of the peatlands of Southeast Asia, highlighting the decrease in vegetation cover due to the increase in agriculture and natural disasters due to climate change. The authors express the need to regenerate peatlands as they are CO2 sinks and serve as ecosystem services.
In methodology, the authors review the publications on peatlands with a high number of works.
In results, most of the publications are excluded because they are not relevant to the topic under study. Of the publications used as relevant, they make a distribution by region and state that the main threats to Indonesia's peatlands are agriculture and palm oil production.
In discussion, the authors say that reforestation projects are being encouraged, and that this study will serve as a basis for the selection of the main services of peatlands.
As a review of the existing publications on peatlands in Indonesia, the study is correct, the authors do an exhaustive job of reviewing the existing publications. However, despite the high number of publications analyzed 149 and 130, the authors do not discriminate those that deal with conservation, they only show a trend in terms of the number of publications and their distribution by region.
This study is interesting, but the authors only conclude that Indonesia's peatlands are threatened by deforestation and agriculture. It is a shame that the authors, with the extensive documentation available, have not carried out a detailed study of the state of conservation. I suggest incorporating a table with the various peatlands and their degree of threat according to the IUCN.
Without knowing the state of conservation, it is not possible to express that this study is the basis for the selection of the main services of the peatlands.
Regarding the 41 references used, they are scarce, however the text speaks of 149 and 130 works consulted.
The article is important because it deals with fragile areas in danger, but it needs to be improved regarding the study of conservation.
Author Response

(The authors gave the same response as above.)

Round 2
Reviewer 2 Report
Comments and Suggestions for Authors
I am satisfied with the author's response and endorse the publication of this manuscript.